

# Comparison of complex networks and tree-based methods of phylogenetic analysis and proposal of a bootstrap method

Aristóteles Góes-Neto[1], Marcelo V.C. Diniz[1], Daniel S. Carvalho[2], Gilberto C. Bomfim[2], Angelo A. Duarte[3], Jerzy A. Brzozowski[4], Thierry C. Petit Lobão[5], Suani T.R. Pinho[6], Charbel N. El-Hani[2,7] and Roberto F.S. Andrade[6]

[1] Department of Microbiology, Universidade Federal de Minas Gerais, Belo Horizonte, Minas Gerais, Brazil
[2] Institute of Biology, Universidade Federal da Bahia, Salvador, Bahia, Brazil
[3] Department of Technology, Universidade Estadual de Feira de Santana, Feira de Santana, Bahia, Brazil
[4] Interdisciplinary Graduate Program in Human Sciences, Federal University of Fronteira Sul, Erechim, Rio Grande do Sul, Brazil
[5] Institute of Mathematics, Universidade Federal da Bahia, Salvador, Bahia, Brazil
[6] Institute of Physics, Universidade Federal da Bahia, Salvador, Bahia, Brazil
[7] National Institute of Science & Technology in Interdisciplinary and Transdisciplinary Studies in Ecology and Evolution (IN-TREE), Instituto de Biologia, Universidade Federal da Bahia, Salvador, Bahia, Brazil

Corresponding author
Aristóteles Góes-Neto, arigoesneto@icb.ufmg.br

## ABSTRACT

Complex networks have been successfully applied to the characterization and modeling of complex systems in several distinct areas of Biological Sciences. Nevertheless, their utilization in phylogenetic analysis still needs to be widely tested, using different molecular data sets and taxonomic groups, and, also, by comparing complex networks approach to current methods in phylogenetic analysis. In this work, we compare all the four main methods of phylogenetic analysis (distance, maximum parsimony, maximum likelihood, and Bayesian) with a complex networks method that has been used to provide a phylogenetic classification based on a large number of protein sequences as those related to the chitin metabolic pathway and ATP-synthase subunits. In order to perform a close comparison to these methods, we selected Basidiomycota fungi as the taxonomic group and used a high-quality, manually curated and characterized database of chitin synthase sequences. This enzymatic protein plays a key role in the synthesis of one of the exclusive features of the fungal cell wall: the presence of chitin. The communities (modules) detected by the complex network method corresponded exactly to the groups retrieved by the phylogenetic inference methods. Additionally, we propose a bootstrap method for the complex network approach. The statistical results we have obtained with this method were also close to those obtained using traditional bootstrap methods.

## INTRODUCTION

The complex networks approach has been successfully applied to uncover organizational principles underlying the origin, evolution, and functioning of several and distinct complex systems in all areas of science, particularly, in the Biological Sciences (*Barábasi & Oltvai, 2004*).

Among the many developed approaches, we call the attention to the phylogenetic analysis based on complex networks using protein or gene similarity as the only source of information. It precludes the use of any previously developed or newly proposed classification model, as required in currently used methods of phylogenetic analysis (distance, maximum parsimony, maximum likelihood, and Bayesian). The basic ideas and essential tools of this approach were presented and thoroughly discussed in *Góes-Neto et al. (2010)* and *Andrade et al. (2011)*. Data of four enzymes present in the chitin metabolic pathway for 1695 organisms (with complete genomes) were collected and used to test and validate the method. This framework was also successfully applied to investigate the evolutionary origins of mitochondria using three ATP synthase subunits and its alphaproteobacterial homologs (*Carvalho et al., 2015*), and to present an evolutionary study of apolipoprotein-E carrying organisms (*Benevides et al., 2016*). Despite these advances, with large databases, we understand this approach should be widely tested, using distinct databases and taxonomic groups, and comparing its results with those provided the current methods quoted above. In order to confer confidence to such a comparison, it is reasonable to use a high-quality database of sequences related to a small number of organisms as well as to propose a bootstrap procedure for the complex network method.

Nowadays, the taxonomy of Basidiomycota fungi is still strongly based in phylogenetic analyses of nucleotide sequences of rRNA genes (18S, 28S) and spacers (ITS) of nuclear ribosomal DNA as well as some protein-coding genes of nuclear origin (*rpb1*, *rpb2*, *tef1-α*) (*Zhao et al., 2017*). However, among the protein-coding genes or their protein products used in phylogenetic reconstructions, none is directly related to exclusive metabolic pathways of fungi, such as that of the biosynthesis of chitin (*Pirovani et al., 2005*). Therefore, phylogenetic inferences based on unique functional fungal proteins are highly desirable.

Chitin, the linear homopolymer of β-1,4-N-acetylglucosamine, is an endogenous structural carbohydrate and one of the main components of the fungal cell wall (*Souza et al., 2009*). In fungi, chitin is synthesized by a pathway containing six steps and the last and irreversible step corresponds to the conversion of UDP-GlcNAc to chitin by the enzyme chitin synthase (E.C. 2.1.4.16) (*Góes-Neto et al., 2010*).

In this study, we conduct further investigations on the reliability of the complex network approach addressing two important issues: (i) we present a direct comparison of its results to those provided by the four phylogenetic methods indicated before; (ii) we propose a bootstrap method that provides a quantitative measure of support values to the branching processes. Although some comparative analysis to other the results of other methods had already been provided, a systematic investigation was missing. On the other hand, a quantitative support for the phylogenetic branches had not been provided up to now. To

this purpose, we have used a high-quality, manually curated and characterized database of chitin synthases of Basidiomycota fungi from representative species.

## METHODS

### Database

The database CHSBasidio was built by data mining using primarily text-based querying in GenPept (NCBI). It comprises complete chitin synthases sequences of Basidiomycota fungi from model species. Each individual enzymatic protein sequence was stored in a single file containing the protein sequence itself and all the relevant associated information, such as indexers, molecular source, structural and functional information, and complete taxonomic classification of the organism from which the sequence was derived, using automatic procedures specifically developed for these tasks.

The knowledge discovery process (KDP) in the CHSBasidio database comprised four steps: (i) data gathering, (ii) screening of collected data, (iii) classification of screened data, and (iv) thorough analyses of classified screened data (*Góes-Neto et al., 2010*). The gathered data were thus screened to eliminate spurious and doubtful (hypothetical or uncertain) reads, and then organized in structured tables with all the relevant associated information.

### Molecular characterization of the CHSBasidio sequences

The complete sequences of the CHSBasidio database were quali-quantitatively meta-analyzed for identification, characterization and comparison at the protein level. The accessions were analyzed, compared and classified mostly using UniProt (*The UniProt Consortium, 2017*), according to the following features: (a) sequence length; (b) theoretical molecular mass; (c) theoretical isoelectric point (p$I$); (d) transmembrane topological organization (transmembrane regions) using TMHMM v.2.0 (*Krogh et al., 2001*); (e) conserved domains (*Marchler-Bauer et al., 2017*); (f) CHS classes (in accordance with the most comprehensive classification recently proposed by *Gonçalves et al., 2016*).

In order to reveal biological patterns in our CHS database and correlate them to the phylogenetic and complex network analyses, we performed a series of univariate, bivariate and multivariate statistical techniques on this customized database. The statistical description of the quantitative univariate protein data (number of entries; smallest, largest and mean values of sequence length, molecular mass and isoelectric point; standard error of the estimate of the mean; variance; standard deviation; median; skewness; and kurtosis), as well as correlation (quantitative data) or association (qualitative data) between all the variables (bivariate analyses) were also performed. A multivariate exploratory method to reveal variation trends (ordination) was also carried out for the complete protein sequences data set. The ordination method of Principal Coordinates Analysis (PCOa) was performed using the Gower index as resemblance measure for the mixed qualitative and quantitative variables. All the aforementioned statistical analyses were performed in PAST 3.0 (*Hammer, Harper & Ryan, 2001*).

### Phylogenetic analyses

The data matrix consisted of 42 sequences from 11 species of basidiomycotan fungi. The sequences were aligned in the most recent version of TCoffee (*Notredame,*
*Higgins & Heringa, 2000*), which aligns proteins by combining the output of many aligners as MAFFT/MUSCLE/PROBCONS/POA/DIALIGN/CLUSTALW/PCMA (http://tcoffee.crg.cat/). The gaps were considered as a 21st character state. Phylogenetic analyses were performed in PAUP 4.0b10 (*Swofford, 2002*), and Mr. Bayes 3.2 (*Ronquist & Huelsenbeck, 2003*), using distance-based (distance matrix) and character-based (maximum parsimony, maximum likelihood and Bayesian) methods.

Mean distances and a neighbor-joining algorithm were used for distance analysis, unweighted parsimony for maximum parsimony analysis. VT (*Muller & Vingron, 2000*) + I (invariable sites) + G (rate heterogeneity among sites: gamma-distributed) best-fit model of protein evolution, previously selected after ProtTest 3.2 (*Abascal, Zardoya & Posada, 2005*), was used for maximum likelihood and Bayesian analyses. Three independent runs were conducted (each with four chains) for $1 \times 10^6$ generations for Bayesian inference.

Clade robustness was assessed using bootstrap proportions (1,000 replicates) for distance, maximum parsimony, and likelihood analyses, and posterior probabilities proportions for Bayesian analysis. Unrooted trees were edited using Geneious v.9. The resulting single distance and maximum likelihood trees and the majority consensus trees of maximum parsimony and Bayesian inference were then subsequently analyzed.

## Network construction and analyses

All networks in this study were constructed based on the similarity degree between amino acid sequences in proteins of the 42 selected organisms. Whenever necessary, one or more of the following indices, which characterize geometrical and topological aspects a given network, were evaluated (*Albert & Barabási, 2002*): node degree ($k_i$), node clustering coefficient ($c_i$), shortest path between two nodes ($d_{ij}$), node betweenness ($b_i^n$), edge betweenness ($b_{ij}^n$), degree assortativity ($q_i$). These indices have local character, as they reflect properties of a given node, of its immediate neighborhood, or of links that are attached to it. Information on the global aspects of the network can be provided by averages of these indices: average degree ($<k>$), network clustering coefficient ($C$), average shortest path ($<d>$), average node betweenness ($B_n$), average edge betweenness ($B_e$), average degree assortativity ($Q$). Other global properties of relevance that have no local counterpart can also be evaluated: network diameter ($D$), probability distribution of nodes with $k$ links ($p(k)$), probability distribution of node clustering coefficients to node degree $k$ ($C(k)$), fractal dimension ($d_b$), modularity ($m_d$). A brief description of the meaning of these indices and functions is provided on the Table 1.

Differently from other methods to construct phylogenetic trees (*Saitou & Nei, 1987*) and phylogenetic networks (*Bryant & Moulton, 2004*), our method is, in fact, a method to detect communities in generally weighted complex networks (*Andrade et al., 2011*). When applied to protein similarity networks, it leads to the phylogenetic classification for the organisms associated to the protein sequences, using less biological assumptions than phylogenetic tree and phylogenetic network methods. The method requires the evaluation of some of these indices, particularly $k_i$, $d_{ij}$, and $b_{ij}^n$. The whole process can be described in the following steps (1–8):

**Table 1 Description of the most usual indices and functions used to characterize a network.** A brief description of the meaning of indices and functions used to characterize the structure of complex networks.

| Symbol | Denomination | Description |
|---|---|---|
| $N$ | Number of nodes | Number of nodes in the network. $N(N-1)/2$ is the maximal number of edges, or connections, the network may contain. |
| $k_i$ | Node degree | Number of neighbors of node $i$, i.e., nodes connected to node $i$ by an edge. |
| $c_i$ | Node clustering coefficient | Number of edges between the neighbors of node $i$ divided by $k_i(k_i-1)/2$. |
| $d_{ij}$ | Shortest path between two nodes | Smallest number of edges necessary to connect two nodes $i$ and $j$. |
| $b_i^n$ | Node betweenness | Number of shortest paths between all pairs of nodes that go through node $i$ divided by $N(N-1)/2$. |
| $b_{ij}^e$ | Edge betweenness | Number of shortest paths between all pairs of nodes that go through node $i$ divided by $N(N-1)/2$. |
| $q_i$ | Degree assortativity | Measure of the average degree of the neighbors of node $i$ as compared to the degree $k_i$. It assumes values between $+1$ and $-1$. The extreme values indicate whether the neighbors of $i$ have degree close or distant from $k_i$. |
| $<k>$ | Average degree | Average value of $k_i$ taken over all $N$ nodes. |
| $C$ | Network clustering coefficient | Average value of $c_i$ taken over all $N$ nodes. |
| $<d>$ | Average shortest path | Average value of $d_{ij}$ taken over all $N$ nodes. |
| $B_n$ | Average node betweenness | Average value of $b_i^n$ taken over all $N$ nodes. |
| $B_e$ | Average edge betweenness | Average value of $b_{ij}^e$ taken over all $N$ nodes. |
| $Q$ | Average degree assortativity | Average value of $q_i$ taken over all $N$ nodes. |
| $D$ | Network diameter | Largest value of $d_{ij}$ |
| $p(k)$ | Probability distribution of nodes with $k$ links | Characterizes one important aspect of network related to the presence or not of hubs (nodes with very large degree in comparison to $<k>$). |
| $C(k)$ | Probability distribution of $c_i$ with respect to $k_i$. | Characterizes the correlation between the values of $k_i$ and $c_i$. |
| $d_b$ | Fractal dimension | Measure of invariance of the distribution of nodes. Relevant when the network admits a cascade of substructures inside similar substructures. |
| $m_d$ | Modularity | Measure of the presence of modules or communities. These are subsets of nodes such that the number of edges between them is much larger than the number of edges linking this subset to other nodes outside it. |

1. *Construction of similarity matrices*: the $n \times n$ similarity matrix ($S$), where $n$ indicates the number of **organisms**, is set up by the comparison of **their** protein sequences using the most recent version of BLAST (*Altschul et al., 1997*). The similarity matrix ($S$) was generated by pairwise alignment, using the score matrix BLOSUM62, which assigns a score for aligning pairs of residues, and determines overall alignment score. The cost to create and extend a gap in alignment is 11 and 1, respectively. The matrix adjustment method that was used to compensate for amino acid composition of sequences was the conditional composition score matrix adjustment (*Altschul et al., 2005*). The elements

$S_{ij}$ of the similarity matrix ($S$) are real numbers in the interval $[0,100]$ corresponding to the percentage of agreement of amino acid types between the two sequences. As BLAST may assign a value $S_{ji}$ different from $S_{ij}$, the matrix $S$ is subsequently symmetrized with its elements being defined by $S_{ij} = \min(S_{ij}, S_{ji})$.

2. *Generation of protein similarity networks*: based on the information stored in $S$, Protein Similarity Networks (PSN) are generated associated with values of a similarity threshold ($\sigma$) according to the following rules: (i) any organism (sequence) is associated to a network node; (ii) a connection is introduced between any pair of nodes ($i,j$) provided $S_{ij} \geq \sigma$. This criterion based on $\sigma$ reflects the evolutionary relationships between the organisms containing the proteins. This strategy makes it possible to replace one single weighted network defined in terms of $\sigma$ by a family of unweighted networks, which can be analyzed by many developed methods and measures (*Albert & Barabási, 2002*; *Newman, 2003*; *Boccaletti et al., 2006*; *Costa et al., 2007*).

3. *Construction of adjacency matrices*: any PSN is represented by its adjacency matrix ($M$), with elements $m_{ij}$ such that: (i) $m_{ij} = 1$, if there is a link between nodes $i$ and $j$; (ii) $m_{ij} = 0$ otherwise. In the current study we generate a set of 101 networks, one for each integer value of the similarity threshold $\sigma \in [0, 100]$.

4. *Construction of neighborhood matrices* $\hat{M}$: for each adjacency matrix $M$, we constructed the set of all neighborhood matrices $M(\ell)$ of order $\ell = 1, \ldots, D$ (*Andrade, Miranda & Lobão, 2006*). Any element $(M(\ell))_{ij}$ is such that it assumes the value 1 only if the shortest path between nodes $i$ and $j$ is $\ell$. Otherwise, $(M(\ell))_{ij} = 0$. Based on the set of $M(\ell)$, a neighborhood matrix $\hat{M}$ is constructed according to:

$$\hat{M} = \sum_{\ell=1}^{D} \ell M(\ell).$$

5. *Distance between networks*: Based on the set of 101 neighborhood matrices, 100 values of the network distance $\delta(\sigma, \sigma + \Delta\sigma)$ between two subsequent networks are evaluated. The network distance $\delta(\alpha, \beta)$ (*Andrade et al., 2008*) is a measure of how two networks $\alpha$ and $\beta$ are distinct from each other. Here, $\delta(\sigma, \sigma + \Delta\sigma)$ indicates the network distance between two PSN evaluated at nearby values $\sigma$ and $\sigma + \Delta\sigma$, with $\Delta\sigma = 1$.

6. *Identification of an optimal value of $\sigma$ using the distances $\delta$*: the set of values $\delta(\sigma, \sigma + \Delta\sigma)$ is characterized by one or more sharp peaks, indicating the threshold values where important changes in the topological structure of the network occur. The value of $\sigma$ where the maximal value of $\delta(\sigma, \sigma + \Delta\sigma)$ occurs defines the critical network and the critical value $\sigma_c$ (*Andrade et al., 2008*). Analyses of the properties of the critical network and of the networks close to $\sigma_c$, as well as those close to the high peaks, are likely to reveal properties related to an evolutionary branching process.

7. *Characterization of the properties of the critical network*: The network indices indicated in beginning of this subsection are evaluated for the networks indicated before, providing auxiliary information on the community structure of the corresponding networks.

8. *Identification of communities in critical networks*: the final step leading to the separation of organisms into phylogenetic groups corresponds to the detection of modules in the networks selected in the step 6. A module is a subset of nodes such that the

number of connections between nodes belonging to the module is proportionally larger than the number of connections between nodes within it to nodes outside it. The Newman–Girvan algorithm (NGA) is used to find the modular structure of the selected networks. It is a bond removal procedure, which amounts to successively removing bonds with largest edge betweenness. At the end of the procedure all bonds are removed, and the community structure analysis results from the dendrogram and the color representation of the renumbered neighborhood matrix (*Andrade et al., 2008*; *Newman & Girvan, 2004*).

## Open software

We developed a software (still at beta version) to evaluate all necessary steps of the community detection procedure according to the complex network method explained in this paper, which is available for download at http://projectsn.github.io/scannet/. The input data is the similarity matrix which, in the current work, was obtained using BLAST. All results reported in the work that are related to this method were obtained using this software. This does not include the steps related to the evaluation of the bootstrap support measures.

## Comparison of complex network method with tree-based phylogenetic methods

The comparison of complex network with tree-based methods of phylogenetic inference for the same dataset and taxonomic group was carried out by: (i) the congruence index $(G(\varphi, \psi) = Q(\varphi, \psi)/R(\varphi, \psi)$, where $R(\varphi, \psi))$: number of common organisms in the critical networks $\varphi$ and $\psi$ and $Q(\varphi, \psi)$: number of organisms in the same community in $\varphi$ and $\psi$ (*Andrade, Pinho & Lobão, 2009*); (ii) topological comparison of the dendrograms generated by each method; (iii) the corresponding metrics (support indexes)—bootstrap (BS) for distance, maximum parsimony, maximum likelihood, and posterior probability (PP) for Bayesian inference; and (iv) the number of removed edges based on the criterion of largest betweenness degree in the complex networks method.

Randomization tests were performed in order to evaluate if there was statistically significant association between the following pair of variables: (i) number of removed edges with largest betweenness degree within the NG procedure and branch length, (ii) number of removed edges during the calculation of betweenness and support indexes of the retrieved groups (BP and PP), using the following parameters: (a) coherence coefficient between variables (resemblance measure) as test criterion ($\lambda$); (b) 1000 iterations, (c) 5% significance level ($\alpha = 0.05$). The randomization tests were carried out in MULTIV (*Pillar, 2001*).

## A bootstrap method for the complex network approach

The bootstrap samples for the complex network method were generated as replicates of the similarity matrix $S$ (see step 1 in section 'Network construction and analyses'). In order to generate a bootstrap sample, each similarity value $S_{ij}$ of the original similarity matrix was divided by 100 and taken as the probability $p$ of success in a binomial distribution. Then, 4,944 samples (this number is the size of the overall alignment for the 42 sequences in our study, including gaps) are drawn at random with replacement from this binomial

distribution, resulting in $k$ successes. Finally, the corresponding entry $S'_{ij}$ in the bootstrap similarity matrix $S'$ is obtained by normalizing the number of successes into a percentile score according to the formula: $S'_{ij} = k \times 100/4{,}944$. Each of the 1,000 similarity matrix bootstrap samples thus generated was then run through the same network-community detection algorithm as the original similarity matrix. The same critical value of $\sigma_c$ as the original similarity matrix (for this study, $\sigma_c = 46\%$) was used for this step. The bootstrap score for any given branch indicates the percentage of bootstrap samples, which featured the community that corresponds to that branch.

## RESULTS AND DISCUSSION

### Biological characterization of CHS database

Our manually curated database comprises 42 unique complete chitin synthase sequences of representative model taxa of the three subphyla of Basidiomycota (Agaricomycotina, Pucciniomycotina, and Ustilaginomycotina), including distinct species with agricultural (the phytopathogens *Moniliophthora perniciosa*, *Puccinia graminis*, and *Ustilago maydis;* and the edible macrofungi, *Agaricus bisporus*, *Flammulina veluticeps*, *Lentinula edodes*, and *Pleurotus ostreatus*) or medical importance (*Filobasidiella neoformans* and *Malassezia pachydermatis*), besides the model species of this phylum (*Coprinopsis cinerea*) (Table 2).

Most of the unique complete protein sequences (76.2%) in the CHSBasidio database comprise proteins that are either predicted (by *in silico* translation) or inferred by homology (when clear orthologs exist in closely related species). Although there are some entries (23.8%) with evidence at the transcript level, with expression data such as the existence of experimentally produced complete cDNA sequences, there is no accession with clear experimental evidence for the existence of the protein by X-ray structure. This can be reasonably explained by the fact that basidiomycotan chitin synthases have few to many transmembrane regions and occur in the cell membrane. These characteristics make the protein crystallization process difficult.

The lengths of the protein sequences range from 519 to 2,066 amino acids residues (mean = 1,135 ± 367.68), but most of the sequences (68%) are in the range from 864 to 1,271 amino acids, exhibiting a marked positive skewness kurtosis. The same is true for the calculated molecular masses of the CHS sequences, which vary from 58.518 to 227.286 KDa (mean = 128.317 ± 45.26), since protein length and molecular mass are highly correlated and precisely linear. The calculated isoelectric points ($pI$) for the enzymes range from 5.42 to 9.30 (mean = 7.72 ± 1.13), but most of the enzymes (71%) are basic. Thus, conversely to protein length and mass, isoelectric point frequency distribution has a marked negative skewness and kurtosis. In fact, both protein length and molecular mass are negatively correlated to isoelectric point, since usually the shorter and lighter the enzymes, the more basic they are (Table 2).

Seven distinct patterns of transmembrane topological organization were identified in basidiomycotan chitin synthases and were named profiles 1–7 (1, 2, 3, 4, 5, 6, 7) (Fig. S1). Profile 2 was the most common (52.4%), followed by profile 1 (26.2%). These two profiles accounted for most of the entries (78.6%). Profile 3 occurred in the minority

Góes-Neto et al. (2018), *PeerJ*, DOI 10.7717/peerj.4349

**Table 2   List of the basidiomycotan chitin synthase sequences used in this work.** List of the basidiomycotan chitin synthase sequences and their corresponding quali-quantitative features and assignment to the groups retrieved in phylogenetic tree-based and network-based analyses.

| Id. No. | Id. Protein (NCBI) | Species | No. amino acid residues | MW (kDa) | pI | Transmembrane regions profile | CDD profile | Conserved domains | CHS class (*Gonçalves et al., 2016*) | Group in phylogenetic analyses | Community in complex network analysis |
|---|---|---|---|---|---|---|---|---|---|---|---|
| 6 | AAW44688 | *Cryptococcus neoformans var. neoformans JEC22* | 947 | 106.385 | 8.86 | 2 | 2 | PF01644 PF08407 | E | A | C3 |
| 7 | AAW42050 | *Cryptococcus neoformans var. neoformans JEC21* | 1,024 | 112.904 | 8.20 | 2 | 2 | PF01644 PF08407 | B | A | C3 |
| 9 | XP_003328707 | *Puccinia graminis f. sp. tritici CRL 75-36-700-3* | 910 | 102.931 | 6.96 | 2 | 2 | PF01644 PF08407 | B | A | C3 |
| 10 | CBQ67873 | *Sporisorium reilianum* | 940 | 104.729 | 7.07 | 2 | 2 | PF01644 PF08407 | B | A | C3 |
| 15 | EFP77369 | *Puccinia graminis f. sp. tritici CRL 75-36-700-3* | 891 | 99.563 | 8.27 | 2 | 2 | PF01644 PF08407 | C | A | C3 |
| 17 | BAJ08815 | *Lentinula edodes* | 866 | 97.134 | 8.37 | 2 | 2 | PF01644 PF08407 | E | A | C3 |
| 19 | EAU80919 | *Coprinopsis cinerea okayama 7 #130* | 941 | 105.442 | 7.61 | 2 | 2 | PF01644 PF08407 | B | A | C3 |
| 24 | ABB70407 | *Puccinia graminis f. sp. tritici* | 919 | 104.076 | 7.84 | 2 | 2 | PF01644 PF08407 | B | A | C3 |
| 29 | BAF76741 | *Pleurotus ostreatus* | 927 | 103.784 | 6.92 | 2 | 2 | PF01644 PF08407 PF03142 | B | A | C3 |
| 31 | BAD20778 | *Malassezia pachydermatis* | 805 | 91.459 | 8.64 | 2 | 2 | PF01644 PF08407 | B | A | C3 |
| 36 | AAW47172 | *Cryptococcus neoformans var. neoformans JEC21* | 996 | 110.026 | 7.26 | 2 | 2 | PF01644 PF08407 | C | A | C3 |

**Table 2** (*continued*)

| Id. No. | Id. Protein (NCBI) | Species | No. amino acid residues | MW (kDa) | pI | Transmembrane regions profile | CDD profile | Conserved domains | CHS class (*Gonçalves et al., 2016*) | Group in phylogenetic analyses | Community in complex network analysis |
|---|---|---|---|---|---|---|---|---|---|---|---|
| 38 | CBQ67884 | *Sporisorium reilianum* | 977 | 108.015 | 6.53 | 2 | 2 | PF01644 PF08407 | C | A | C3 |
| 41 | ADX07309 | *Flammulina velutipes* | 1,676 | 185.073 | 6.00 | 7 | 2 | PF01644 PF08407 PF03142 | C | A | C3 |
| 4 | CAB96110 | *Agaricus bisporus* | 909 | 102.259 | 8.45 | 2 | 2 | PF01644 PF08407 | III | B | C2 |
| 5 | AAW43575 | *Cryptococcus neoformans var. neoformans JEC21* | 931 | 104.441 | 8.38 | 2 | 2 | PF01644 PF08407 | III | B | C2 |
| 11 | EFP91815 | *Puccinia graminis f. sp. tritici CRL 75-36-700-3* | 870 | 97.036 | 8.97 | 2 | 2 | PF01644 PF08407 | III | B | C2 |
| 16 | EFP76086 | *Puccinia graminis f. sp. tritici CRL 75-36-700-3* | 969 | 108.438 | 7.03 | 2 | 2 | PF01644 PF08407 | III | B | C2 |
| 23 | ABB70409 | *Puccinia graminis f. sp. tritici* | 977 | 109.334 | 6.90 | 2 | 2 | PF01644 PF08407 | III | B | C2 |
| 25 | ABB70408 | *Puccinia graminis f. sp. tritici* | 868 | 96.632 | 8.85 | 2 | 2 | PF01644 PF08407 | III | B | C2 |
| 28 | BAF37219 | *Pleurotus ostreatus* | 938 | 105.202 | 8.62 | 2 | 2 | PF01644 PF08407 | III | B | C2 |
| 30 | ABW09311 | *Moniliophthora perniciosa* | 913 | 102.762 | 8.68 | 2 | 2 | PF01644 PF08407 | III | B | C2 |
| 33 | XP_570882 | *Cryptococcus neoformans var. neoformans JEC21* | 755 | 85.383 | 8.74 | 2 | 2 | PF01644 PF08407 | III | B | C2 |
| 40 | ADX07313 | *Flammulina velutipes* | 620 | 70.162 | 9.30 | 6 | 2 | PF01644 PF08407 PF03142 | III | B | C2 |
| 42 | ADX07293 | *Flammulina velutipes* | 864 | 97.184 | 7.57 | 2 | 2 | PF01644 PF08407 PF03142 | III | B | C2 |

Peer J

**Table 2** (*continued*)

| Id. No. | Id. Protein (NCBI) | Species | No. amino acid residues | MW (kDa) | pI | Transmembrane regions profile | CDD profile | Conserved domains | CHS class (*Gonçalves et al., 2016*) | Group in phylogenetic analyses | Community in complex network analysis |
|---|---|---|---|---|---|---|---|---|---|---|---|
| 1 | XP_566840 | *Cryptococcus neoformans var. neoformans JEC21* | 1,271 | 143.041 | 5.46 | 1 | 1 | PF03142 | Vb | C | C1 |
| 2 | AAB71697 | *Cryptococcus neoformans var. grubii* | 1,041 | 116.201 | 8.84 | 1 | 1 | PF03142 | IVb | C | C1 |
| 3 | BAC78196 | *Coprinopsis cinerea* | 1,409 | 157.676 | 8.92 | 1 | 1 | PF03142 | IVa | C | C1 |
| 8 | XP_003328148 | *Puccinia graminis f. sp. tritici CRL 75-36-700-3* | 1,417 | 155.072 | 7.41 | 4 | 1 | PF03142 | IVa | C | C1 |
| 12 | EFP89079 | *Puccinia graminis f. sp. tritici CRL 75-36-700-3* | 1,729 | 192.445 | 6.28 | 3 | 3 | PF03142 PF00063 | Va | C | C1 |
| 13 | EFP83544 | *Puccinia graminis f. sp. tritici CRL 75-36-700-3* | 1,212 | 135.277 | 8.28 | 1 | 1 | PF03142 | IVb | C | C1 |
| 14 | EFP78527 | *Puccinia graminis f. sp. tritici* CRL 75-36-700-3 | 2,066 | 227.286 | 5.63 | 3 | 3 | PF03142 PF00063 | Vb | C | C1 |
| 18 | XP_001830485 | *Coprinopsis cinerea okayama7#130* | 1,147 | 125.865 | 8.77 | 1 | 1 | PF03142 | IVb | C | C1 |
| 20 | AAB84284 | *Ustilago maydis* | 1,486 | 162.724 | 9.00 | 1 | 1 | PF03142 | IVa | C | C1 |
| 21 | AAB84285 | *Ustilago maydis* | 1,180 | 130.623 | 6.07 | 1 | 1 | PF03142 | Vb | C | C1 |

Góes-Neto et al. (2018), *PeerJ*, DOI 10.7717/peerj.4349

**Table 2** (*continued*)

| Id. No. | Id. Protein (NCBI) | Species | No. amino acid residues | MW (kDa) | pI | Transmembrane regions profile | CDD profile | Conserved domains | CHS class (*Gonçalves et al., 2016*) | Group in phylogenetic analyses | Community in complex network analysis |
|---------|-------------------|---------|------------------------|----------|-----|------------------------------|-------------|-------------------|------------------------------------|-------------------------------|--------------------------------------|
| 22 | ABB70406 | *Puccinia graminis f. sp. tritici* | 1,019 | 113.685 | 9.02 | 1 | 1 | PF03142 | IVb | C | C1 |
| 26 | ABB70410 | *Puccinia graminis* | 1,997 | 222.514 | 6.44 | 3 | 3 | PF03142 PF00063 | Va | C | C1 |
| 27 | BAF41225 | *Pleurotus ostreatus* | 1,436 | 159.927 | 8.44 | 1 | 1 | PF03142 | IVa | C | C1 |
| 32 | BAF37218 | *Lentinula edodes* | 1,937 | 215.318 | 5.42 | 3 | 3 | PF03142 PF00063 | Vb | C | C1 |
| 34 | AAW45092 | *Cryptococcus neoformans var. neoformans JEC21* | 1,423 | 158.113 | 7.85 | 1 | 1 | PF03142 | IVa | C | C1 |
| 35 | AAW44838 | *Cryptococcus neoformans var. neoformans JEC21* | 1,236 | 136.325 | 8.77 | 1 | 1 | PF03142 | IVb | C | C1 |
| 37 | AAW44187 | *Cryptococcus neoformans var. neoformans JEC21* | 1,895 | 214.398 | 6.41 | 3 | 3 | PF03142 PF00063 | Va | C | C1 |
| 39 | XP_003327390 | *Puccinia graminis f. sp. tritici CRL 75-36-700-3* | 519 | 58.518 | 7.16 | 5 | 1 | PF03142 | IVb | C | C1 |

of the accessions, and the four others were unique: each one occurred for only one sequence. Profile 1 comprised six transmembrane helices distributed as followed: two in the N-terminal region, one in the medial region, and three in the C-terminal region of the protein; Profile 2 encompassed seven transmembrane helices in the C-terminal region; Profile 3 were represented by six transmembrane helices, one in the medial, one in sub-terminal and three other in the C-terminal part of the protein. Profile 4 resembled to some degree the profile 1, as well as profiles 5 and 6 did regarding to profile 2. Profile 7 was the most divergent, exhibiting six transmembrane helices, all in the medial region of the protein (Table 2).

Three distinct patterns of types of conserved domains were identified in the CHSBasidio database and were designated as profiles 1, 2, and 3. Profile 1 corresponded to the sole presence of the conserved domain PF03142. Profile 2 comprised those entries with the conserved domains PF01644 and PF08407. Profile 3 included the accessions containing the conserved domains PF03142 and PF00063. Profile 2 was the most frequent in the sequences (57.1%). Profile 1 appeared in 31% of the entries while profile 3 was found in only 11.9% of the complete CHS sequences (Table 2). Furthermore, there is a statistically significant ($p = 1.5016E–7$) correlation (Spearman coefficient = 0.71) between identified transmembrane topological organization and conserved domains profiles. This probably occurred because some of the conserved domains also include transmembrane helices.

There are many distinct and, mainly, contradictory classifications of chitin synthase (CHS) isoenzymes in different fungal groups. Recently, *Gonçalves et al. (2016)* performed a genome-wide analysis in more than 800 putative chitin synthases in proteomes associated with about 130 complete genomes of all known evolutionary lineages of organisms. This large-scale analysis not only allowed the authors to completely revise and unify the fungal CHS classification but also to develop an excellent searchable, web-based reference database (http://wwwabi.snv.jussieu.fr/public/CHSdb/).

*Gonçalves et al. (2016)* classified basidiomycotan CHS in eight classes: B, C, E, III, IVa, IVb, Va, Vb. The classes B, C and E are some subdivisions of the classical class II in the older fungal CHS classifications (*Niño Vega, Carrero & San-Blas, 2004*). All these aforementioned classes for basidiomycotan CHS, according to *Gonçalves et al. (2016)*, were represented in our CHSBasidio database.

Principal Coordinates Analysis (PCoA) was based on five variables (sequence length, isoelectric point, transmembrane topological organization profile, conserved domains profile, and CHS class according to *Gonçalves et al., 2016*) for the 42 sampling units (entries). The basidiomycotan CHS sequences were ordinated based on the first two coordinates (coord. 1: 38.23% and coord. 2: 27.69%), which jointly represented approximately 66% of the explained variation according to their qualiquantitative attributes. All classes Va and Vb CHS sequences located in the positive portion of the coordinate 1 and mostly above this coordinate. All classes IVa and IVb CHS sequences occurred in the negative portion of the Coordinate 2 and mostly, in the negative portion of coordinate 1. Conversely, all CHS classes III, B, C and E located in the positive portion of coordinate 2, with class III sequences in the less positive portion and B, C, and E in the more positive portion of this coordinate 2 (Fig. 1). Thus, PCO feature-based approach

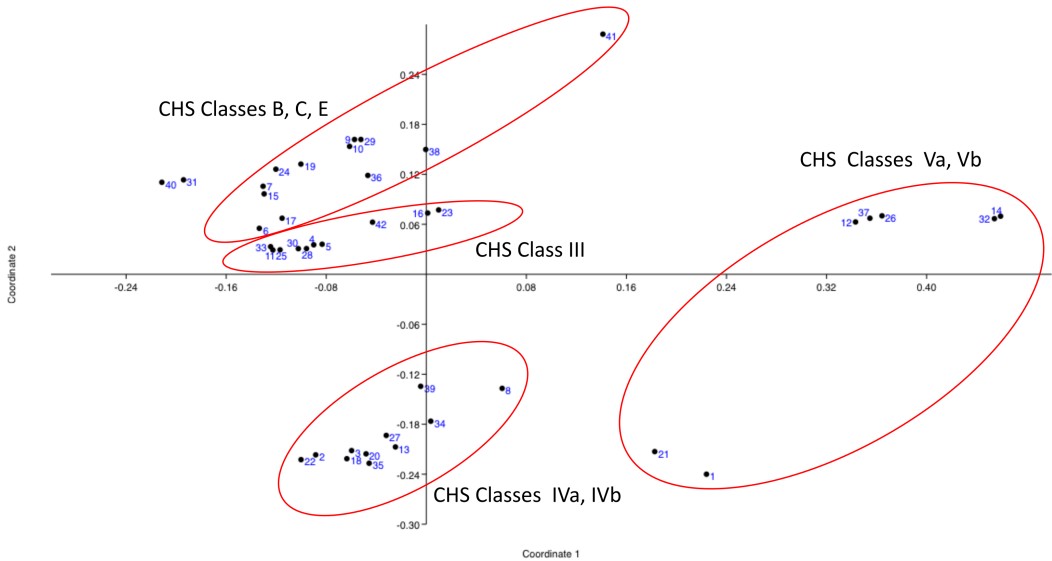

**Figure 1** **Principal Coordinates Analysis (PCOa) of the qualiquantitaive feature-based matrix of complete sequences of proteins of CHSBasidio database.** The basidiomycotan CHS sequences were ordinated based on the first two coordinates (coord. 1: 38.23% and coord. 2: 27.69%), which jointly represented approximately 66% of the explained variation according to their qualiquantitative attributes.

analysis is in complete accordance with the most detailed and large-scale classification of chitin synthases of all fungi until date (*Gonçalves et al., 2016*).

## Tree-based phylogenetic methods
### *Maximum parsimony*

The matrix of aligned sequences of basidiomycotan chitin synthases was 4,944 characters long (including gaps), of which 63% were variable and 58.4% of the variable characters were parsimony informative. The final result of the parsimony analysis comprised only one most parsimonious unrooted tree, which showed three distinct clades (named A, B and C), besides a more inclusive clade composed by the union of A and B clades (named AB) (Fig. 2). The first dichotomous division was observed between the most inclusive clade AB and clade C. The most inclusive clade AB showed a 100% bootstrap value and, thus, it was maximally supported while for clade C a low support value of only 50% bootstrap was obtained. Clades A and B, despite being topologically supported by *bootstrap* values above 50%, showed very contrasting values: clade B, similarly to the most inclusive clade AB, exhibited 100% bootstrap, while for clade A, topological confidence was a few above half (54%).

### *Distance*

The final result of distance analysis comprised only one unrooted tree. The unrooted tree showed, as well as in parsimony analysis, the same distinct three groups (named as A, B and C), besides a more inclusive group formed by the union of groups A and B (named as AB) (Fig. 2). The most inclusive group AB exhibited, similarly to maximum parsimony analysis, maximum support index, with a 100% bootstrap value, while group C also had

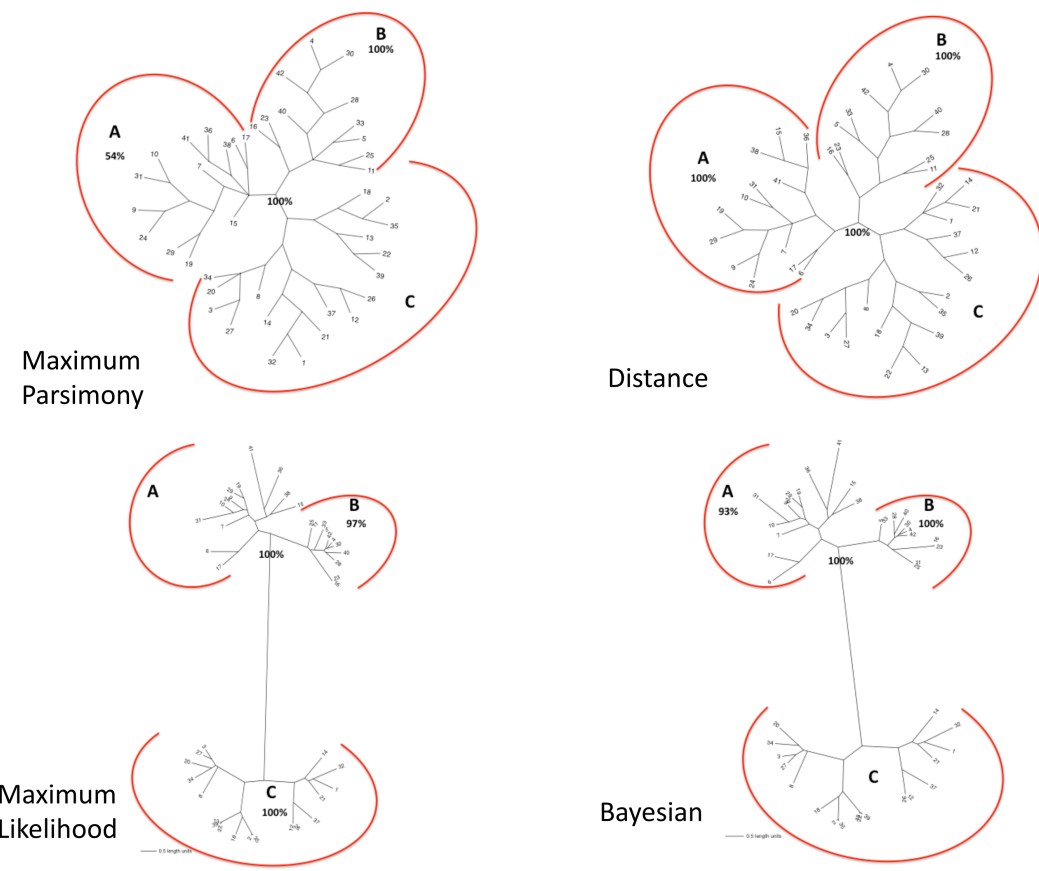

**Figure 2** (A) **Maximum Parsimony majority-rule consensus tree;** (B) **Distance tree;** (C) **Maximum Likelihood tree;** (D) **Bayesian majority-rule consensus tree.** Trees of the fourtree-based phylogenetic methods. Bootstrap values above 50% are exhibited. Scale bar represents the number of amino acid substitutions.

low support (<50% bootstrap). Both groups A and B had also maximum values of support, with 100% bootstrap. Conversely to what was retrieved in maximum parsimony analysis, the topological confidence of group A was significantly higher.

### Maximum likelihood

The unrooted tree retrieved the same which was obtained in maximum parsimony and distance analyses: the same three distinct groups (named A, B and C), besides a more inclusive group composed by the union of A and B (named as AB) (Fig. 2). The most inclusive group AB showed, similarly to maximum parsimony and distance analysis, the maximum value of bootstrap (100%), but, conversely to those analyses, group C, as well group AB, also exhibited maximum support value (bootstrap = 100%). Group B, individually, also showed a value next to maximum support but slightly lower (97% bootstrap) than those found in maximum parsimony and distance analyses. Nevertheless, in marked contrast to those analyses, group A had a low support value (<50% bootstrap).

**Table 3** Comparison of the support indexes (bootstrap and posterior probabilities) of the tree-based methods.

| Groups | Maximum parsimony (% BP) | Distance (% BP) | Maximum likelihood (% BP) | Bayesian (% PP) |
|--------|--------------------------|-----------------|---------------------------|-----------------|
| AB | 100 | 100 | 100 | 100 |
| C | <50 | <50 | 100 | <50 |
| A | 54 | 100 | <50 | 93 |
| B | 100 | 100 | 97 | 100 |

### Bayesian

The final result of Bayesian analysis was a majority consensus unrooted tree. Once again, it was retrieved the same as obtained in the analysis of maximum parsimony, distance and maximum likelihood, with the same three distinct groups (named as A, B and C), besides the more inclusive group formed by the union of groups A and B (named as AB) (Fig. 2). The most inclusive group AB showed, similarly to the maximum parsimony, distance and maximum likelihood analyses, the maximum support value (100% posterior probability), but, conversely to the results obtained in maximum likelihood and similarly to those obtained in maximum parsimony and distance analyses, group C exhibited a low support value (posterior probability <50%). Group B, as in maximum parsimony and distance analyses, also exhibited the maximum support value, 100% posterior probability. Nevertheless, similarly to distance analysis and differently from maximum parsimony and maximum likelihood, group A had a significantly high support, with 93% posterior probability.

Table 3 comparatively summarizes the support indexes of bootstrap (BP) and posterior probability (PP), retrieved from the four tree-based phylogenetic analysis methods for the studied database (non-congruent indexes amongst different methods are in red).

In all the tree-based phylogenetic analyses (distance, maximum parsimony, maximum likelihood and Bayesian), the group A comprised only basidiomycotan CHS isoenzymes of the classes B, C and E, while group B encompassed exclusively CHS of the class III, and group C consisted of all CHS of the classes IVa, IVb, Va and Vb. Moreover, the group AB, retrieved in all phylogenetic analyses corresponded exactly to the join grouping of classes B, C, E and III CHS sequences. Therefore, all the four phylogenetic methods retrieved exactly the fungal CHS classification of *Gonçalves et al. (2016)* for the Phylum Basidiomycota.

### Complex networks method

The optimum value of similarity (critical similarity) used to retrieve the phylogenetic relations (*Andrade et al., 2011*) was $\sigma_c = 46\%$ (Fig. 3). Near this critical value, there is an abrupt topological change with the disaggregation of a sole network completely connected in groups that can be discerned, with maximum phylogenetically-relevant information in relation to noise, enabling the detection of communities (modules) in the network (Fig. 4).

The selected critical network that best represented the phylogenetic relationships of the studied dataset has order $(N) = 42$ nodes and size $(M) = 198$ edges.

This critical network has the following indexes in relation to:

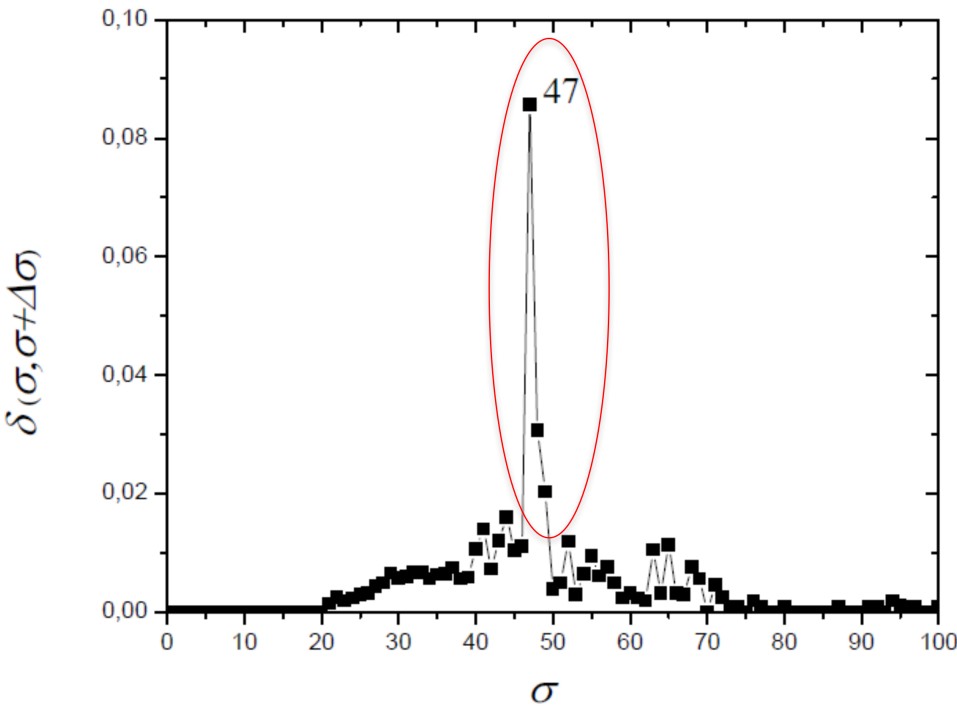

**Figure 3 The distance $\delta(\sigma, \sigma + \Delta\sigma)$ between networks for successive similarities at the maximal value with $\Delta\sigma = 1$.** The optimum value of similarity (critical similarity) used to retrieve the phylogenetic relations (*Andrade et al., 2011*).

I. Connectivity: (i) the degrees of nodes varies from 1 to 13 and the average degree is $<k> = 9.429$; (ii) the probability distribution of nodes with degree $k$ corresponds to a bimodal distribution: 7 nodes with low value of $k$ centered in $k = 3$, and 35 nodes with high values of $k$ centered in $k = 11$;

II. Assortativity $Q = 0.00216$;

III. Distance: (iv) average minimal path is $<d> = 3.272$, (v) diameter $D = 7$, (vi) average node betweenness $B_n = 137.4$ (vii) average edge betweenness $B_e = 3.27$;

IV. Cluster: (viii) average clustering coefficient $C = 0.815$; mainly in the interval $[0.6, 1.0]$, only 5 nodes with $c_i$ value below 0.6;

V. Auto-similarity: (ix) fractal dimension $d_b = 1.37$;

VI. Modularity $m_d = 0.975$;

The detection of modular structure (that is, the identification of communities in the critical network $\sigma_{crit} = 46\%$) was carried out by joint analysis of the color representation of the neighborhood matrix (Fig. 5) along with the dendrogram generated by successive link elimination according to the betweenness index (using edge betweenness) (Fig. 6).

The critical network $\sigma_{crit} = 46\%$ exhibited three very distinct communities (modules), named C1, C2, and C3 (Fig. 7). The community C2 was the one with highest connectivity, as one can see in the color matrix.

Communities C2 and C3 showed more connected between them, with four nodes with a high edge betwenness value (C2: 16 and 23; C3: 19 and 29) and four edges with high

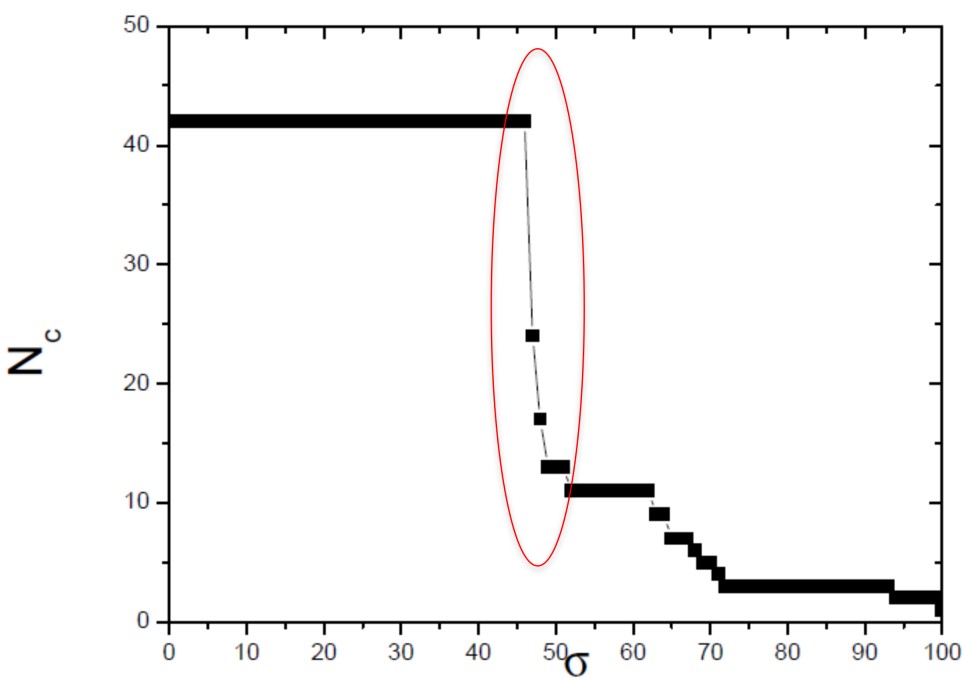

**Figure 4** **Size of the largest connected component (Nc) versus the threshold similarity $\sigma$.** Near the critical value, there is an abrupt topological change with the disaggregation of a sole network completely connected in groups that can be discerned, with maximum phylogenetically-relevant information in relation to noise, enabling the detection of communities (modules) in the network.

edge betweenness (inter-communities: 29–23, 29–16; 19–23, 19–16) that connect them. Community C1, however, had only one inter-community edge, linking the vertices C1:12 and C3:38.

## Comparison of the complex networks method and traditional tree-based methods of phylogenetic analyses

The classifications generated by the complex networks method and the four tree-based methods of phylogenetic analysis (distance, maximum parsimony, maximum likelihood, Bayesian) exhibited maximum congruence index: $G(\varphi, \psi) = 100\%$, with $\varphi$, $\psi$ corresponding to a community CX generated by any two methods. The communities C1, C2 and C3, detected by the complex networks method, corresponded exactly to the groups C, B and A obtained through the phylogenetic inference methods, respectively, and, thus, the critical network ($\sigma_{crit} = 46\%$) retrieved all the phylogenetic relationships of the basidiomycotan CHS. The closer relationship of groups A and B, forming the more inclusive group AB, was also detected by the complex networks method, through the higher number of vertices connected by inter-community edges of high betweenness between C2 and C1. Furthermore, additional phylogenetic relationships, which cannot be directly visualized by a tree graph, were also evidenced as exhibited by the intercommunitary edges between nodes in communities C2 and C3, and between communities C1 and C3 (Table 2 and Fig. 7).

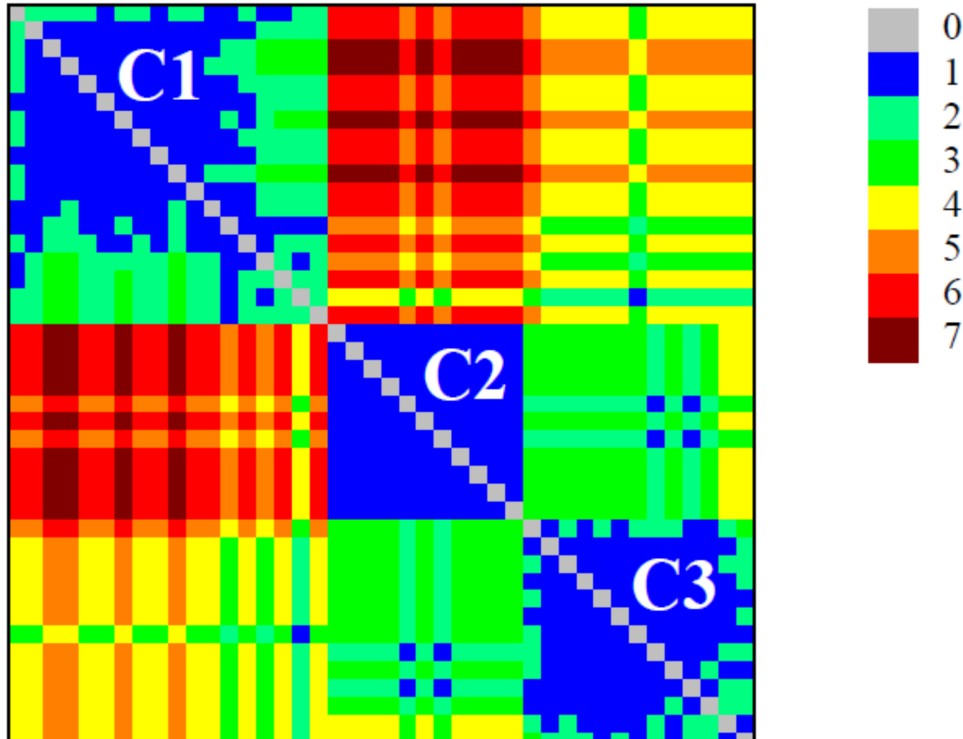

**Figure 5** **Colour plot of neighbourhood matrix at $\sigma_{cri} = 46\%$ with the indication of the communities (modules).** The detection of modular structure (that is, the identification of communities in the critical network) was carried out by joint analysis of the color representation of the neighborhood matrix.

Table 4 synthesizes, in a comparative manner, the indices of number of removed edges (in the complex networks method) and the support indexes of bootstrap (BP) and posterior probability (PP) (in the maximum parsimony, distance, maximum likelihood, and Bayesian methods) for the main retrieved groups/communities. However, a statistically significant correlation between the number of removed edges by the complex networks method and the support measures of the traditional methods of phylogenetic analysis was not detected.

The dendrograms generated by tree-based phylogenetic methods are a graphic representation of evolutionary relationships between included organisms or their molecules, such as proteins (*Russo, Miyaki & Pereira, 2012*). The trees are generated based on the assumption that the evolutionary process is strictly divergent but the widespread existence of reticulated evolution (horizontal/lateral transfers, hybridizations, and non-dicothomous cladogenesis) implies that the evolutionary process can be concurrently divergent and convergent so that the best graphic representation would not be a tree (*Kunin et al., 2005*). Therefore, a more realistic graphical representation of phylogenetic hypotheses can be generated by a complex networks method (*Bapteste et al., 2013*). Tree graphs are a subset of general graphs or networks. Trees are optimized visualizations of often much more complex evolutionary signals. Using trees, additional dimensions of information in the data are overlooked (*Bapteste et al., 2012*). The scope of our evolutionary
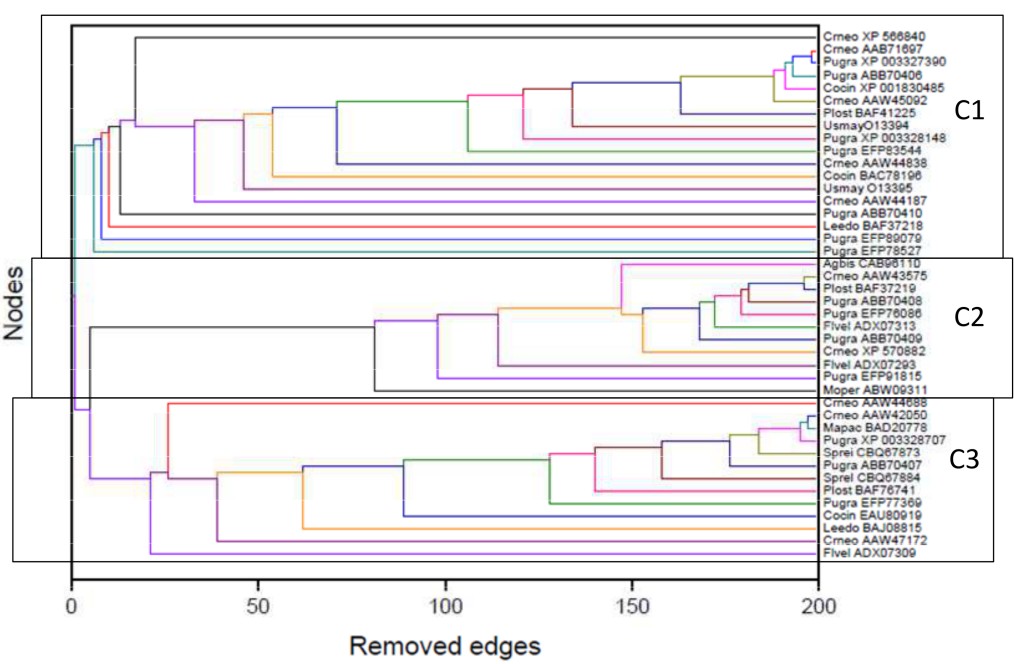

**Figure 6** **Dendrogram produced by the successive elimination of edges with the largest value of betweenness.** The detection of modular structure, that is, the identification of communities in the critical network was carried out by joint analysis of the color representation of the neighborhood matrix (Fig. 5) along with the dendrogram generated by successive link elimination according to the betweenness index.

thinking should be moved beyond a tree-thinking to a network-thinking paradigm (*Bapteste et al., 2013*).

## Comments on the used bootstrap method

The topologies generated by the phylogenetic analyses must be submitted to some confidence test that quantitatively evaluates the statistical support of the inferred or proposed groups (*Felsenstein, 2004*). The percentage values of bootstrap (distance, maximum parsimony, maximum likelihood, Bayesian) and posterior probability (Bayesian) are the most used support measures: the closer to the maximum value, the more robust is the retrieved topology; that is, the more reliable is the formed group/community. Using the bootstrap method based on random resampling over similarity scores, we have obtained results that were strikingly similar to those used for the tree-based methods. Notably, the same groups A, B and C as in the tree-based methods were also retrieved by the complex network method, with bootstrap values of 100%, 100%, and 74.6%, respectively. In this scenario, we also found a bootstrap support of 100% for group AB (Fig. 8).

This bootstrap procedure is distinct from those used in the tree-based methods since it does not rely on the resampling of characters (*Felsenstein, 2004*); instead, the locus of the resampling procedure is here the similarity matrix, which serves as the starting point for the community detection algorithm. The rationale behind this procedure is that it may emulate to some degree the consequences on the similarity matrix that would be obtained if one were to resample characters (i.e., amino acids) directly, since this would

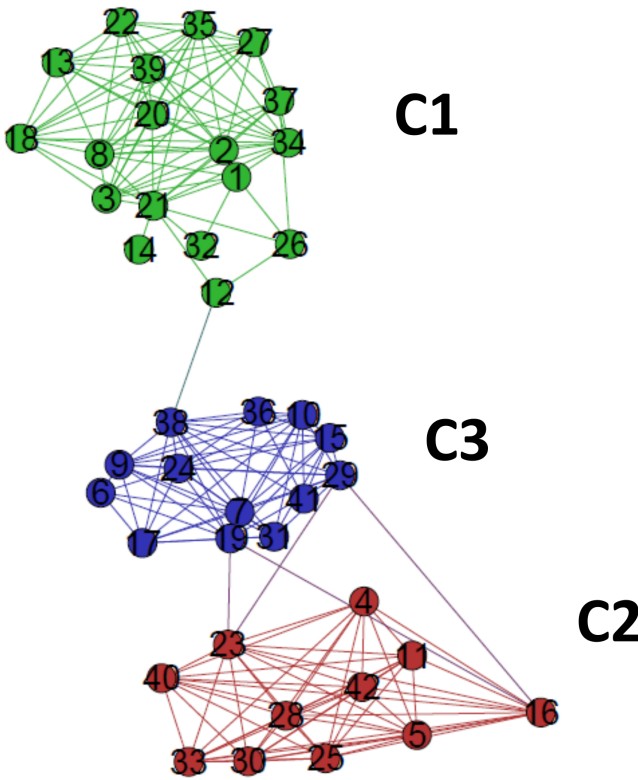

**Figure 7  The standard network representation at $\sigma_{cri} = 46\%$ (Gephi) with the indication of the communities (modules).** The critical network exhibited three very distinct communities (modules), named C1, C2, and C3.

**Table 4  Comparison of the number of removed edges and support indexes (bootstrap and posterior probabilities) of tree-based and complex networks methods.**

| Groups | Complex networks | Maximum parsimony | Distance | Maximum likelihood | Bayesian |
|---|---|---|---|---|---|
| A + B = C3 + C2 | 5 | 100 | 100 | 100 | 100 |
| C = C1 | 6 | <50 | <50 | 100 | <50 |
| A = C3 | 21 | 54 | 100 | <50 | 93 |
| B = C2 | 75 | 100 | 100 | 100 | 100 |

plausibly result in a variance on the similarity scores that would depend on the similarity scores themselves. That is, by resampling characters, we could expect a peak of variance in similarity scores across bootstrap samples around the similarity score of 50%, since at this point there would be a greater margin for the alignment of bootstrap sequences to either improve or worsen over the original sequences. We also expect the variance for bootstrap similarity scores to decrease as we approach either end of the similarity spectrum. The method of resampling over a binomial distribution herein adopted emulates this behavior. Across our bootstrap samples, the highest variance (0.66) was found in one of the entries

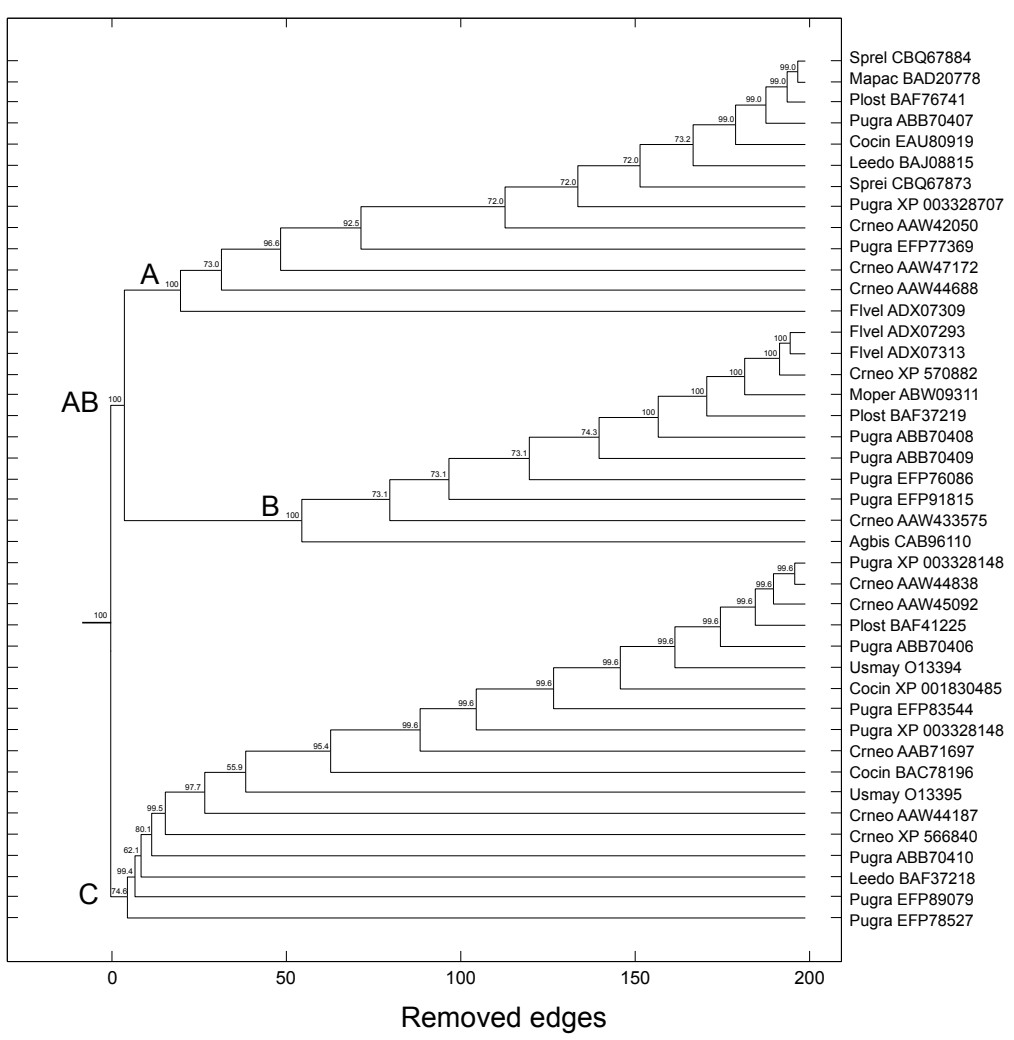

**Figure 8** **Dendrogram produced by the successive elimination of edges with the largest value of betweenness.** Bootstrap values for all branches were produced according to the method described in the text. Bootstrap support for communities AB (C3 + C2), A (C3), B (C2), and C (C1) in complex network method. (Note: the terminology A, B and C is those used in tree-based methods).

with an original similarity score of 46, while an entry with an original similarity score of 95 exhibited a variance of 0.11 (Table S1).

It should be noted that, by performing a bootstrap resampling according to the method here described, each bootstrap sample results in an adjacency matrix $m$ (see section 'Network construction and analyses', step 3, above) that is slightly different than the one obtained from the original similarity matrix and $\sigma_c$. We also performed tests in which the bootstrap samples were generated by direct random rewiring of the adjacency matrix itself, yielding similar results to the method adopted herein. However, the rewiring method demands additional statistical assumptions for which we are not ready to provide support at this point.

It may be objected that this bootstrap method yields scores that cannot be compared to those in the tree-based methods. Strictly speaking, however, bootstrap scores for the complex network-based method are calculated in exactly the same way as in the other methods; what differs is the resampling method. While traditional resampling from sequences, followed by the application of the network-based method, can in principle be done for each bootstrap sample, in practice this is computationally much more demanding than resampling from the similarity matrix. The degree to which those methods are equivalent will be the subject of a future study.

In complex networks, the statistical measure that could be used in a similar manner as the support measures used in phylogenetic analyses (BP and PP) is the number of removed edges during the calculation of betweenness. When one removes edges with high betweenness, a great perturbation in the system is caused, which can imply in the rupture of network structure, and the more resilient to attacks a community/group is, the more robust it is (*Costa et al., 2007*). In this case study with chitin synthases of Basidiomycota fungi, the community C2 (= group B in phylogenetic analyses) was the one that exhibited the highest value when considering this statistical measure. Nonetheless, the lack of statistically significant correlation between a probable support measure for communities in complex networks and the traditionally used support measures in phylogenetic methods (BP and PP) involving a same dataset suggests the need of a more comprehensive investigation about this topic. Currently, our research group has been generating and analyzing many complex networks with high order and size to answer this fundamental question.

## Concluding remarks

In this work, we compared a complex networks method with the traditional methods of phylogenetic analysis (distance, maximum parsimony, maximum likelihood, and Bayesian), using a manually curated and characterized database of chitin synthases of Basidiomycota fungi from model species. The three modules detected by the complex networks method corresponded exactly to the groups retrieved by the aforementioned phylogenetic inference methods. By applying the method again to the values of $\sigma = 52$, 63 and 65, which locate the three small secondary peaks in the Fig. 4, we were able to provide a finer sub-community analysis. The intra-community links that survive in the presented analyses correspond to adjacency matrix elements $m_{ij} = 1$ and similarity score $S_{ij} > 46$, which are about to be erased at these higher threshold values. Furthermore, we proposed and successfully tested, for the first time, a bootstrap method, whose results were close to those obtained using traditional bootstrap in current phylogenetic methods.

Finally, we remark that complex network formalism can be applied to investigation subsequent transitions that give rise to sub-communities within C1, C2 and C3. To this purpose we use the secondary peaks of $\delta(\sigma, \sigma + \Delta\sigma)$ identified at $\sigma = 52$, 63, and 65%. Then, the same sequence of steps (2–8) indicated previously at these values $\sigma$ to obtain dendrograms and neighborhood matrices uncovering the subcommunity structure.

## ACKNOWLEDGEMENTS

We thank all who contributed directly or indirectly to this work, especially the CNPq (Conselho Nacional de Desenvolvimento Científico e Tecnológico) and the Graduate Programs of Microbiology and Bioinformatics of the Universidade Federal de Minas Gerais (UFMG) for infrastructure and for partially financing this study, and also, Jose Mario Vicensi Grzybowski for help with the computational implementation of the community-detection algorithm.

### Funding

Roberto F.S. Andrade received a productivity in research grant from the National Council for Scientific and Technological Development (CNPq), Brazil (no. 305060/2015-5) (http://www.cnpq.br). Thierry Petit Lobão received a productivity in research grant from the National Council for Scientific and Technological Development (CNPq), Brazil (no. 307140/2009-1) (http://www.cnpq.br). Aristóteles Góes-Neto received a productivity in research grant from the National Council for Scientific and Technological Development (CNPq), Brazil (no. 310764/2016-5) (http://www.cnpq.br). Suani T.R. Pinho received a productivity in research grant from the National Council for Scientific and Technological Development (CNPq), Brazil (no. 306458/2015-2) (http://www.cnpq.br). Charbel N. El-Hani received a productivity in research grant from the National Council for Scientific and Technological Development (CNPq), Brazil (no. 301259/2010-0) (http://www.cnpq.br). We are supported by PRONEX/FAPESB-CNPQ, INCTI-SC and INCT-CITECS programs. The funders had no role in study design, data collection and analysis, decision to publish, or preparation of the manuscript.

### Grant Disclosures

The following grant information was disclosed by the authors:
National Council for Scientific and Technological Development (CNPq): 305060/2015-5, 307140/2009-1, 310764/2016-5, 306458/2015-2, 301259/2010-0.
PRONEX/FAPESB-CNPQ.
INCTI-SC.
INCT-CITECS.

### Competing Interests

The authors declare there are no competing interests.

### Author Contributions

- Aristóteles Góes-Neto, Jerzy A. Brzozowski and Roberto F. S. Andrade conceived and designed the experiments, performed the experiments, analyzed the data, contributed reagents/materials/analysis tools, wrote the paper, prepared figures and/or tables, reviewed drafts of the paper.

- Marcelo V.C. Diniz, Daniel S. Carvalho and Gilberto C. Bomfim performed the experiments, analyzed the data, reviewed drafts of the paper.
- Angelo A. Duarte performed the experiments, analyzed the data, contributed reagents/materials/analysis tools, reviewed drafts of the paper.
- Thierry C. Petit Lobão, Suani T.R. Pinho and Charbel N. El-Hani conceived and designed the experiments, analyzed the data, contributed reagents/materials/analysis tools, wrote the paper, reviewed drafts of the paper.

## Data Availability

The patterns of transmembrane topological organization of CHS sequences are provided in Fig. S1.

## Supplemental Information

Supplemental information for this article can be found online at http://dx.doi.org/10.7717/peerj.4349#supplemental-information.

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
