# Peer review of "Comparison of complex networks and tree-based methods of phylogenetic analysis and proposal of a bootstrap method"

_PeerJ, doi:10.7717/peerj.4349_

## Round 0.1 · original submission · Major Revisions

Two of three reviewers do not recommend that this paper is published here and all three reviewers have provided constructive criticisms that are largely in agreement. If the authors believe that they can address these major criticisms, I would be happy to read a revised version of the manuscript.

The presentation of the paper in particular background, motivation, central contribution and technical aspects detailed by reviewers 1 and 2 need to be addressed. All reviewers express concern that the manuscript does not adequately place this effort in the larger context. Although it is beyond the scope of a single manuscript to convince the community of the value of network methods, addressing the criticisms is an important step forward.

The choice of chitin as a target here is brought up by almost all reviewers. Although it is perhaps relatively standard now to use "synthetic data" (where we know the correct answer), this manuscript uses a single example to showcase their method. The limitations of this should be discussed clearly in your manuscript. The comment of reviewer #3 regarding whether simpler methods could not be used with equal efficacy is important, and that comparison should be made if possible.

·

Basic reporting

Overall the English was clear and unambiguous. However, there were many terms that I was unfamiliar with that need in-text explanation (beyond citation to a reference), specifically: "betweeness", "assortativity", "auto-similarity", “removed edges”, and "modularity". This is an interesting and new method, and therefore the authors need to take cognisance that this will be interesting and new to many readers - please help us with explanations readily at hand in the text.

There are many many different types of networks. I strongly encourage the authors to briefly set-out in the Introduction how this type relates to the other types - even if it is to say that this is completely different (then how is it different). There is a lot of confusion out there about networks, and so you want to highlight to the reader that this is a new type (not some knock-off adjustment of, for example, NeighbourNet). This does not need to be an extensive review of network types, but just an alert to the reader about how your method is different.

Another angle on this is why and when should researchers use this method over the others - it strikes me that this method is good at picking up groups when there are very deep phylogenetic divergences. Other methods, like NeighbourNet, usually fall apart (or rather descend into spider-web black hole) when there are deep divergences and homoplasies are present. This point bugged me for quite a while reading the manuscript - it was only when I ran a few additional network analyses did the benefits of your method become clear (I do have to put a proviso that my analyses were handicapped by the use of the similarity matrix, and I'm not entirely clear as to how this was calculated).

The raw data has been shared, although the similarity matrix lacks taxon labels - this makes it difficult to assess the results of this study with other analyses (for example, I ran a NeighbourNet on the distance matrix but could not relate it to the final figures as there were no labels). In addition, the phylogenetic tree analyses (tree and bootstrap tree sample) have not been included – please add.

Experimental design

I have a few concerns that need to be addressed.
1. Similarity Calculation
The authors use a protein sequences to establish the relationships between sub-phyla of Basidiomycota fungi. I have not used protein sequences before, but it seems that there has been substantial variation in the selected region/s across the lineages. The alignment of these protein sequences is very messy - for example, the protein sequences range from 519 to 2066 amino acids residues. I have a concern about how this affects the calculation of the similarity matrix. The first step of the complex networks approach is the construction of a similarity matrix and this is the basis for all subsequent steps. The authors state that the used a BLAST search. This does not explain how gaps are dealt with, are similarities calculated pairwise or across all samples (i.e. gaps are deleted across the entire alignment). As the way that the distances are caculated can dramatically influence the distance values, this needs to be clearly explained.
For example, the distance matrix calculated by PAUP for the NJ analyses was likely very different fom the distance matrix produced for the first step of the complex network calculation. I am pretty sure of this as when I generate a NJ tree based on the author-supplied distance matrix, it does not have the three clear clades as shown in Fig. 3b. Thus, is comparing the NJ tree and the network results a fair comparison as it is not an algorithm superiority, but rather how gaps are dealt with between the different methods.

2. Lack of support for clade C in phylogenetic analyses: a rerooting artefact
Trees were rerooted in FigTree. This is a common mistake and should NEVER be done. Support values are for branches, but in the NEWICK format are stored at nodes. By rerooting the relationship between nodes and branches is changed and thus the support values can be shifted up or down onto a different branch! Given the very long branch leading to clade C in all analyses, it should be well-supported (just from eye-balling, but medium levels of homoplasy could also be driving this). Rooting has to be done on each bootstrap sample and on the representative tree before combining this information. I am against rooting in this setting anyway – it is unlikely to actually capture the correct ancestor-descendent relationship. Rather pesent unrooted trees. Even better would be to show the consenses networks of the bootstap sample (Holland et al. 2004 - Using consensus networks to visualize contradictory evidence for species phylogeny) – although most phylogeneticist still seem to find this heretical.
3. Software
What software was used to conduct the complex network analyses – i.e. if I wanted to use this analysis, I don’t think I could use you methods to figure out how to do this.
4. Complex networks testing
Something for the authors to consider for another paper – simulate some data under a range of scenarios, from tree-like to non-tree-like datasets, and demonstrate how the algorithm performs. Even better would be to use datasets developed for testing other networks.

Validity of the findings

The complex network approach detects the same clades as found in the phylogenetic trees. As I suspect that once the rerooting issue is sorted, all clades will have high phylogenetic support, then the authors have demonstrated that the method detects the same clades. What now? How is this an improvement on the current state? The power of networks is that they can show non-bifurcating patterns for further explanation of evolution (or conflicting pattern in the dataset). However, the authors do not address the intra-community connections – what do these connections mean? How should we interpret them? The lines between taxa seem to simply indicate a relationship – but what kind of relationship? This needs to be discussed.

Additional comments

This is an interesting manuscript, but I was at times scratching my head – not through any error in the manuscript, but through my reasonable ignorance (it is a “new” approach afterall). Please make some effort to provide more “road signs” explaining the terms and reasoning that you followed.
Some minor in-text comments:
Line 39: “disclosure” - this does not seem to be the right word.
L44: Remove “have”
L46: Remove “have”
L90: Please explain “evidence of existence”
L94-101: I found this section very confusing – please include a short introduction as to why you did this.
L95: “smallest, largest, mean values” of what?
L133-137: What was the threshold criterion?
L152: The symbols for the math notation have been removed and replaced with squares.
L160: Please describe the “community detection” a little bit more.
L190: This concept of “removed edges” is used fairly often, but I don’t get it. Please find somewhere to provide a brief explanation of the thinking behind this.
L282: What is a “symmetric” matrix of aligned sequences?
L282: Previous stated max length of sequences was ~2000, now the alignment is ~5000. Dealing with gaps, yes? Please make this clear...
L288-290: What is the point in reporting this information – it is in the Figure.
L313: Odd sentence, meaning unclear.
L398-406: This is the only section on network interpretation, and it is quite vague. Please expand and make relevant in light of your network results.
L416: Is this method actually bootstrapping. It strikes me to more along the lines of Monte Carlo resampling.
L430: A range of alpha values is selected to compare network support values with those of the trees. The alpha values affect the network support values – so how would one go about selecting an alpha value in the absence of tree information...
Table 3: Again, I don’t understand this concept of “removed edges”
Fig 2: The qualiquantitative analysis seems to be a loose thread in the manuscript – it is not clearly related back to the phylogenetic or network results. Please highlight more clearly to the reader why this was an important analysis.
Fig 5&6: As these share the same x-axis, consider merging into Fig 5a and 5b.

Reviewer 2 ·

Basic reporting

In this manuscript, Goes-Neto and collaborators apply network methods they have been developing over the past few years to a dataset of 42 chitin synthases from Fungi, in order to produce a phylogeny of these sequences, and then compare this phylogeny to ones built with more classical methods (parsimony, NJ, ML and Bayesian). They also improve their previous protocol and implement robustness measures by ‘rewiring’ the adjacency matrices that are used for their networks. Professional English is used throughout the text, and introduction and background are sufficient. Literature could be improved in some parts though. Especially, using networks to infer phylogenies has been suggested by others researchers before 2010, for instance by Atkinson et al. (doi: 10.1371/journal.pone.0004345). Manuscript structure conforms to PeerJ standards, raw data are adequately supplied, and figures are generally good (although resolution should be improved on Fig 7).

Experimental design

My main concern with this manuscript lies with the research question that is being investigated. In this paper. The authors show that network methods seem (see next paragraph) to be able to reconstruct phylogenies with results that are comparable to more standard (and time-consuming) methods. But their results only concern one dataset (chitin synthases) with relatively low phylogenetic diversity (Fungi), and there is absolutely no reason to believe that their methods will perform as well on other datasets. As much as I am convinced that network methods should definitely be more used by the evolutionist community (and especially the phylogeneticists), I strongly doubt that this paper will convince an already somehow reluctant audience. In order to do so (and again, I’d really like to see that happen), one would need a very thorough benchmarking of the method on simulated data, so that at least the true tree is known, which is not the case here. Chitin synthases evolution within Fungi is actively investigated, and the authors should definitely compare their results with published results (see for instance Goncalves et al., 2016, doi: 10.1186/s12862-016-0815-9). Moreover, benchmarking should be performed under conditions that are notoriously difficult for more classical phylogenetic methods: different tree shapes (balanced/unbalanced), different evolutionary rates (slow/fast), different sequence lengths (short/long), different sequence composition (unbiased/biased), and most of all varying relative rates (molecular clock evolution/non molecular clock evolution). From my experience, fungal chitin synthases are not even an especially difficult dataset for phylogenetic methods, which somehow renders the results less convincing. I understand this is a whole research program in itself, but, as is, I’m afraid this manuscript will not convince evolutionary biologists to use network methods for inferring phylogenies.
Unfortunately, the Materials and Methods part is very difficult to read, partly because it relies on notions that were previously published by the same team, but are not explained in the present manuscript. More problematically, some explanations are also missing. The only equation (line 152) somehow didn’t make it to the PDF I received, so I cannot assess what it means. Also, the network distance delta (line 155) is not explained. Network properties (lines 162-168) are given as is, and surely will not mean anything to the average evolutionist. Finally, the most prominent problem is that the actual method to build the phylogeny from the networks (lines 180-195) is hardly explained (but only provided as a reference to a previous article from the team), as is the newest part of the method (rewiring of the matrices).

Validity of the findings

As explained before, my main concern lies within the experimental design, but sadly the findings are not without flaws either. The authors chose to root their trees using midpoint rooting, but such a method only works with ultrametric distances, which biologically translates to sequences evolving under molecular clock (i.e. all sequences have accumulated the same amount of substitutions). Molecular clock however is generally not achieved for large time scales (such as the timespan elapsed since the ancestor of Basidiomycota for instance). Sadly, since the authors chose not to display branch lengths on their trees, one cannot know whether their data are actually under molecular clock, although it’s very likely they are not (again, see Goncalves et al.). As a result, rootings (and hence clades) are most likely devoid of biological meaning, which renders the following analyses much less relevant. Besides, there are much better ways to compare trees than just comparing whether the major ’clades’ actually correspond or not. Robinson-Foulds distances are classically used for such a purpose and it would be nice to actually have that kind of measures in the manuscript.

Additional comments

Again, I’d like to state that I’d really like for the authors to publish their methods, but in order to do so they’ll have to convince the phylogeneticists community, which for the moment is a bit wary of networks. For this reason, the authors might want to be a bit more thorough on the phylogenetic analyses and on the biological terms they use. For instance, building an NJ tree from mean BLAST distances (line 110) is definitely not the canonical method. Lots of more biological distances are available for such purposes (see the choice that is offered by any phylogenetic software for instance). Also, the term ‘midterm’ rooting’ (line 119) is never used, as is ‘terminals’ (line 288). Similarly, branch support values in Bayesian analyses are posterior probabilities, not bootstrap values (line 334). More generally, bootstrap has a very precise meaning in phylogenetics (i.e. building replicates of the initial character matrix by resampling sites), so the rewiring method that the authors propose should probably not be called ‘bootstrap’, but rather ‘robustness measures’. Finally, the manuscript is overly long in many places, and should be trimmed down in order to be more legible.
As a conclusion, I’m afraid I cannot recommend publication for this manuscript in PeerJ.

Reviewer 3 ·

Basic reporting

line 110 - was there any correction for multiple hits appplied?

line 136 - does this simply mean m_ij = 1 if S_ij > some threshold? If so, what is the threshold? Ooops nevermind, this is answered in step 3. It seems like steps 2 and 3 should be merged in the description.

line 152 is not intelligible in my pdf of the manuscript


page 13 "calculus" -> "calculation"

"These aforementioned characteristics strongly difficult the protein crystallization processs, which is a mandatory step before X-ray diffraction analysis." ->
"These characteristics make the protein crystallization process difficult"

Experimental design

The procedure for producing the sequence alignment needs to be described.

Validity of the findings

No comment

Additional comments

My background is in phylogenetics rather than machine learning. So perhaps, I am biased. Nevertheless, I have a hard time viewing this paper as an advance. The authors show that for the case of a hand-chosen chitin synthase dataset. Their method of building complex networks with random rewiring is able to recover the 3 major genealogical groups that are recovered by phylogenetic methods. However, the groups were recoverable using a very simple PCA analysis, so this does not seem to be a challenging task.

But the paper fails to explain why this is an advance. It is not clear at all how general the result is, and it is not clear to the reader why we need another method to identify the features of the data that seem to be easy to detect using any phylogenetic method.

I just didn't understand the point of the paper. The method (including the new form of bootstrapping) seems to lack any statistical interpretation. So it is very unclear to me why someone would use this method.

---

## Round 0.2 · accepted · Accept

We respectfully ask that you carefully go through the English of your paper closely and remove typos etc while in production.